# Modification of Poly(lactic acid) with Orange Peel Powder as Biodegradable Composite

**DOI:** 10.3390/polym14194126

**Published:** 2022-10-02

**Authors:** Nonni Soraya Sambudi, Wai Yi Lin, Noorfidza Yub Harun, Dhani Mutiari

**Affiliations:** 1Department of Chemical Engineering, Universitas Pertamina, Simprug, Jakarta 12220, Indonesia; 2Department of Chemical Engineering, Universiti Teknologi PETRONAS, Seri Iskandar 32610, Perak, Malaysia; 3Department of Architecture, Universitas Muhammadiyah Surakarta, Jl. A Yani, Mendungan, Kartasura 57169, Indonesia

**Keywords:** poly(lactic acid), orange peel powder, biodegradation, tensile strength, Young’s modulus, biocomposites

## Abstract

Traditional fossil-based plastic usage and disposal has been one of the largest environmental concerns due to its non-biodegradable nature and high energy consumption during the manufacturing process. Poly(lactic acid) (PLA) as a renewable polymer derived from natural sources with properties comparable to classical plastics and low environmental cost has gained much attention as a safer alternative. Abundantly generated orange peel waste is rich in valuable components and there is still limited study on the potential uses of orange peel waste in reinforcing the PLA matrix. In this study, orange peel fine powder (OPP) synthesized from dried orange peel waste was added into PLA solution. PLA/OPP solutions at different OPP loadings, i.e., 0, 10, 20, 40, and 60 wt% were then casted out as thin films through solution casting method. Fourier-transform infrared spectroscopy (FTIR) analysis has shown that the OPP is incorporated into the PLA matrix, with OH groups and C=C stretching from OPP can be observed in the spectra. Tensile test results have reviewed that the addition of OPP has decreased the tensile strength and Young’s modulus of PLA, but significantly improve the elongation at break by 49 to 737%. Water contact angle analysis shows that hydrophilic OPP has modified the surface hydrophobicity of PLA with a contact angle ranging from 70.12° to 88.18°, but higher loadings lead to decrease of surface energy. It is proven that addition of OPP improves the biodegradability of PLA, where PLA/60 wt% OPP composite shows the best biodegradation performance after 28 days with 60.43% weight loss. Lastly, all PLA/OPP composites have better absorption in alkaline solution.

## 1. Introduction

Plastic is a durable, inexpensive, and lightweight polymer that has a broad range of applications ranging from food packaging to electronic goods. Over 381 million tonnes of plastic are produced annually, and it was reported that around 55% was for single-use purpose only, 25% was incinerated, and 20% was recycled [1]. The overwhelming usage of conventional petroleum-based plastic which is highly non-biodegradable leads to environmental challenges including pollution and depletion of non-renewable resources [2]. The manufacturing process of plastic products consumes intensive energy, while the common ways of disposing plastic waste such as landfill, incineration and dumping into the oceans have led to adverse impacts on the environment [3,4]. Shrinking land capacity is one of the consequences of plastic waste spread in the natural environment as plastic products usually take up to several hundreds of years to decompose completely in landfills [5,6]. It was also estimated that, up to 14.5 metric tonnes of plastic ends up in the ocean per year, causing irreversible impacts on marine life and contamination of water sources [7]. Disposing plastic wastes through incineration could significantly reduce its volume by 80 to 95% [3]. However, burning of plastics releases toxic heavy metal and harmful gases like dioxins and methane into the atmosphere [3,8].

Due to the increasing concerns on plastic pollution, biopolymer has attracted extensive interest in recent studies [9,10]. The environmental-friendly biopolymer derived from renewable sources or being able to decompose naturally are preferred as safer alternatives [11,12,13,14]. Agricultural crop-based feedstocks and biomass waste are the common sources in the production of bioplastics [3]. The former, is however, less favourable due to its competition for land, water sources, and food production [3]. Poly(lactic acid) (PLA), also known as Polylactide, is a renewable polymer synthesized through esterification of lactic acid derived from sugar fermentation [15,16]. It can be derived from agricultural sources such as corn starch, wheat, and sugar cane [3]. Being viewed as the mostly potentially biopolymer to replace the conventional petroleum-derived plastic, PLA received significant attention due to its excellent properties, including biocompatibility [3], good mechanical properties and processability into various products, as well as low carbon footprint [17]. However, drawbacks such as high brittleness and low barrier properties have limited some of the applications of PLA [17]. Under controlled and specific industrial composting conditions, PLA can be degraded within a few days or up to several months, whereas it can take up to hundreds of years to decompose in landfill [6].

Furthermore, citrus waste disposal has been challenging as it is generated abundantly but underutilized. According to Raimondo et al., the volume of citrus fruits processed is nearly 31.2 million tons per year, while 50 to 60% of the original mass become the residue [18,19]. The disposal of citrus waste in landfill remains unsatisfactory as its low pH may cause soil salinity as well as other harmful effects to ruminants [19]. Among the citrus waste, orange peel is rich in valuable components such as pectin and cellulose fibres. Pectin with good gelling ability and cellulose fibres that give strength could act as new building block in products [18], while cellulose is naturally degradable by microbial activities [19]. These benefits of orange peel waste have attracted attention for their potential uses in reinforcing the conventional petrochemical or biobased matrices [19]. The incorporation of orange peel powder into renewable PLA is believed can enhance the biodegradability of the biobased polymer while improving its mechanical properties such as flexibility [6]. Bassani et al. conducted research on the incorporation of antioxidant extract from orange peels to develop an innovative PLA-based active packaging [17]. Phenolic compounds in orange peels are one of the valuable active materials that exhibits good antioxidant property [20]. However, the study regarding orange peel modification in PLA matrix limits to its extract, and so far didn’t address the mechanical strength, surface properties and biodegradability of composites. Hence, there is still need of study in combining orange peel powder with PLA to form biocomposites.

In this study, orange peel powder is incorporated into pure PLA to improve its biodegradability in natural environment while maintaining its good physical properties. Orange peel waste without prior chemical modification is converted into fine powder through mechanical grinding, then mixed at different concentrations with PLA in chloroform solution. The homogeneous solution is then casted onto flat surface to obtain smooth composite films through solution casting method. The biodegradability of the biocomposite film is studied by conducting soil burial degradation experiment over 28 days.

## 2. Materials and Methods

### 2.1. Materials

PLA (IngeoTM biopolymer 3251D) was purchased from NatureWorks LLC (Plymouth, MN, USA). Sweet orange (Citrus sinensis) was purchased from local supermarket in Seri Iskandar, Perak. Chloroform (CAS 67–66–3) was purchased from R&M Chemicals (Edmonton, AL, Canada). Hydrochloric acid (HCl) (CAS 7647–01–0) was purchased from Thermo Fisher Scientific (Waltham, MA, USA). Sodium hydroxide (NaOH) was purchased from Sigma Aldrich (St. Louis, MO, USA). Deionized (DI) water was utilized throughout the experiment. The chemicals are utilized without further purification or treatment.

### 2.2. Pre-Treatment and Synthesis of Orange Peel Powder

Sweet orange (Citrus sinensis) peel collected was washed with tap water to remove impurities on the surface. Orange peel was dried under the sun for 20 h, followed by drying in oven for 18 h at 60 °C according to previous method by Farahmandfar [21]. Dried orange peel was then reduced into smaller size using pestle and mortar, then milled into fine powder using electric blender. The powder was sieved using sieve shaker to get uniform powder size of 100 μm, then stored in plastic container at room temperature until use.

### 2.3. Synthesis of Composite Films

PLA pellets were dissolved in chloroform under constant stirring for 1 h at room temperature to make 10 wt% PLA solution. Around 0.5 g orange peel powder (OPP) was added into the PLA solution and stirred for another 1 h to prepare PLA/10 wt% OPP solution. The solution was casted onto flat glass plate and four petri dishes, respectively and dried for 24 h at room temperature, which is similar to experiment done by Musa et al., in synthesis of PLA/banana fibres film [22]. The procedures were repeated by varying the amount of orange peel powder to prepare PLA/OPP solutions containing 20 wt%, 40 wt%, and 60 wt% OPP. Previous studies have shown that high loading of cellulose increased the strength and stiffness of composite [23,24,25].

### 2.4. Characterization of Composite Films

The morphology of film was captured by using stereoscopic microscope (Leica S8 APO). The sample functional groups were characterized using Fourier Transform Infrared Spectroscopy (FTIR) Perkin Elmer Spectrum One. Blank PLA sample of size 2 × 2 cm^2^ was placed on the FTIR. The intensity of transmitted light was computed in a wave range of 4000–500 cm^−1^. For water contact angle analysis, same size PLA film was mounted on goniometer. A single drop of deionized water was released from the syringe on top onto the specimen surface. The angle between the specimen surface and the edge of the water drop was measured. The mechanical strength analysis by using universal testing machine (UTM, Instron, Norwood, MA, USA). Blank PLA film was prepared into sample of dumbbell shape following the ASTM D638 standard [26]. Geometry of the sample, including width, gauge length, and thickness were recorded into the software, while sample was loaded onto the grip of 5 kN Universal Testing Machine. The grip separation was started and run at a constant speed of 5 mm/min until sample break. Tensile strength (MPa), Young’s Modulus (MPa), and elongation at break (%) of the sample were analysed from graph generated.

### 2.5. Swelling Test

Three samples with size of 4 × 4 cm^2^ were prepared from each composite and weighed to record respective initial weight as W_dry_. Samples were immersed into three petri dishes containing different pH solutions, i.e., 0.1 M HCl, distilled water, and 0.1 M NaOH, respectively [27]. After 1 h, samples were removed from the solutions and liquid droplets on the surface were wiped off. The weight after immersion was recorded as W_swollen_. The swelling index was calculated Equation (1) as below
(1)$$\mathrm{Swelling}\;\mathrm{index}\;(\%)=\frac{W_{\mathrm{swollen}}{\;-\;W}_{\mathrm{dry}}}{W_{\mathrm{dry}}}\;\times \;100\%$$

### 2.6. Biodegradability Test

Blank PLA and different concentrations of PLA/OPP composite samples casted in petri dish of diameter 8 cm were weighed to measure respective initial weight (W_1_). Each sample was buried under 2 cm of moist garden soil in a container and kept for 28 days in an open environment. The soil burial method is adapted from previous studies by Park et al., and Lertphirun et al., with some modification [28,29]. The soil was kept moist by spraying sufficient amount of water. Degradation of the samples was observed at an interval of 7 days for continuous 4 weeks period. The sample residues were collected from soil, rinsed with water to remove the soil on its surface, followed by drying for 30 min at 40 °C in oven. Weight after (W_2_) of samples were recorded. The biodegradability of the composites was measured using Equation (2) as follows:(2)$$\mathrm{Biodegradability}\;(\%)=\frac{W_{2\;}{-\;W}_{1}}{W_{1}}\;\times \;100\%\;$$

## 3. Results

### 3.1. Morphology, Functional Groups and Contact Angle

Blank PLA film (Figure 1a) retained its high transparency property and displayed a smooth, shiny surface. As OPP was introduced at low loadings, i.e., 10 and 20 wt%, the composites (Figure 1b,c) turned slightly yellowish but maintained characteristics of blank PLA where the film still appeared as a shiny and transparent surface. More distinct difference was observed when the OPP′s loadings increased to 40 and 60 wt%, the composite appeared as bright orange films and lost its surface shininess. Furthermore, increased roughness was detected on the composites′ surfaces due to the increased concentration of OPP.

FTIR characterization was conducted to determine the functional groups present in the blank PLA matrix and after incorporation of OPP. Figure 2 shows the FTIR spectra ranging from wavenumber 4000 cm^−1^ to 550 cm^−1^. The small peaks at 2946.09 to 2996.32 cm^−1^ are attributed to the CH stretching vibration, and sharp absorption peaks at 1749.29 cm^−1^ belongs to the C=O stretching which can be observed in all samples [30,31].

Water contact angle between water droplet and the composite film′s surface was measured to study its hydrophobicity or known as surface wettability. Bar chart in Figure 3 illustrates the water contact angle (θ) obtained for each film. The blank PLA film shows an angle of 73.82°, corresponding to its hydrophobic nature which has a static contact angle within the range of 60 to 85° [32]. This value is also close to the measured angle of blank PLA film (67.27°) prepared using same solution casting method as reported by Alakrach et al. [33].

### 3.2. Mechanical Properties

It has been observed that the incorporation of OPP in the PLA matrix has weakened its mechanical properties, i.e., tensile strength and young’s modulus decrease when compared to the blank PLA film (Figure 4). The initial tensile and modulus exhibited by blank PLA film are 13.21 MPa and 1646 MPa, respectively. The PLA/OPP composites of all loadings display lower tensile strength within the range of 2.59 to 6.9 MPa, and lower modulus within the range of 145 to 652 MPa. However, there is a significant improvement on the elongation at break. PLA/20 wt% OPP, PLA/40 wt% OPP, and PLA/60 wt% OPP composites show an improved elongation (15.32, 9.98, and 10.6 mm) compared to blank PLA film (2.73 mm).

### 3.3. Swelling

Swelling rate (%) is calculated by comparing the weight before and after immersing each composite sample in acid, alkaline, and neutral distilled water at pH 1, 13, and 7, respectively. Based on Figure 5, it is observed that the samples′ swelling percentage generally increases with increased OPP loadings, regardless of the pH of the solution. The blank PLA film shows the lowest swelling rate among all samples, which is corresponding to its hydrophobic nature [29]. A significant increase in the swelling percentage can be observed in PLA/10 wt%, PLA/20 wt%, PLA/40 wt%, and PLA/60 wt% composites.

### 3.4. Degradation

Soil burial experiment was conducted for a total period of 28 days, where the samples were taken out, weighed, and observed at every 7-day interval. Figure 6 shows the graph of weight loss percentage of each film against days after soil burial, while Table 1 presents and compares the films′ appearance before and after soil burial.

The blank PLA film has shown the least weight loss throughout the soil burial experiment, with low percentage ranging from 3.73 to 5.49%. From Table 1, it can also be observed that the blank PLA film experiences very little changes in appearance due to its hydrophobic nature and slow degradation rate. Similar result was obtained by Lertphirun and Srikulkit [29] for blank PLA degradation study in soil burial test. With increased loadings of OPP in PLA, the composites show an increase trend of weight loss percentage and degradation. The highest weight loss is observed in PLA/60 wt% OPP composite that shows a weight loss percentage in the range of 25.03 to 60.43%, which is 8 to 12 times higher than that of blank PLA film. Moreover, it can be observed from Table 1 that cracks, holes and fungi formation appear on the surface of PLA/10 wt% OPP, PLA/20 wt% OPP, PLA/40 wt% and PLA/60 wt% OPP composites and they have become more brittle after soil burial. These features increase by time for all composites while more obvious changes are clearly visible in higher loadings PLA/OPP composites.

## 4. Discussion

As observed from Figure 1b,c, good and even dispersion of OPP particles in PLA matrix was achieved at relatively low loadings at 10 and 20 wt%. When OPP loading increased to 40 wt%, it could be clearly seen that the OPP particles no longer distributed evenly on the surface, agglomeration and some aggregates were observed especially at the top right corner of the film (Figure 1d). At the highest concentration of 60 wt% OPP, the whole composite has become denser, with very little or no space in between could be observed (Figure 1e). For FTIR spectra in Figure 2, a broad peak in the high energy region (2944.72 to 3309.94 cm^−1^) indicates the presence of hydroxyl group, which is only observed at PLA/OPP composites at higher loadings [31]. This is due to the large amount of OH groups present in carbohydrates and lignin in the orange peel [31,34,35], and is more visible when the OPP loadings are high as the fibre content is increased. Similar trend is observed for the C=C stretching at 1608 cm^−1^ due to the presence of unsaturated aromatic compounds in orange peel [34,35]. Lastly, the peaks at 1081.31 to 1181.32 cm^−1^ indicates the presence of C-O stretching in all spectra [36,37]. Addition of hydrophilic filler OPP which is rich in cellulose fibres has slightly modified the hydrophobic nature of PLA. PLA/10 wt% OPP composite reports a decreased contact angle (70.12°), which means that it is now more hydrophilic. The presence of OH groups in OPP increases the polarity and surface energy of the composite film [33]. High surface energy creates a strong attractive force which make water droplet to spread out better on the surface, thus showing a good wettability. However, at 20, 40, and 60 wt% OPP loadings, water contact angle increases to 78.4°, 81.72°, and 88.18°, respectively, showing that the composites have again shifted towards hydrophobic nature. This is because of the high loadings of OPP in PLA matrix increases the surface roughness, which subsequently reduces the surface energy [33], causing the composites to become slightly hydrophobic.

It is believed that the cellulose components in OPP have helped in modifying the toughness by improving the elongation. The poor interfacial adhesion between cellulose fibres and polymer can cause agglomerations due to hydrophilic nature of OPP and hydrophobic nature of PLA, hence resulting in low tensile and modulus [38,39]. While addition of OPP can seem to overcome the brittleness of PLA by the increasing elongation at break (Figure 4c). Similar reduction in tensile strength and increasing in elongation at break were observed in previous study by modifying PLA with cellulose as fillers [31,40,41], where the addition of cellulose can interlock with PLA matrix and ease the movement of composite upon stress, hence increasing the elongation.

Furthermore, the optimum loading is obtained at 20 wt% OPP that yields the highest elongation at break which gives the polymer a good ductility to not break easily. Nonetheless, at even higher OPP loadings (40 wt% and 60 wt%), all three mechanical properties, i.e., tensile strength, Young’s modulus, and elongation at break are found decreased again. This is probably due to the high OPP loadings that cause poor distribution and agglomeration of the filler particles, resulting in embrittlement of the composites. The mechanical strength results show similar agreement to study by Singh et al., whereas high loading of cellulose in PLA reduced tensile strength and modulus, while elongation at break was increased [42]. The orange peel powder might act as plasticizer, to improve the ductility of material, as can be seen by the increasing of elongation at break at around 49 to 737%, usually by sacrificing tensile strength and stiffness [42]. The imperfect distribution of orange peel powder and weak interfacial adhesion between orange peel powder and PLA can be other factors that contribute to the lowering tensile strength and modulus [43].

The addition of OPP with hydrophilic cellulose fibre content has helped to bond the water molecules together through the affinity of OH groups, resulting in higher absorption or swelling rate [44,45]. Comparing all three solutions of different pH, the composites experience the highest swelling rate in alkaline NaOH, with the maximum rate 80.22% achieved by PLA/60 wt% OPP composite. Moreover, cracking was observed after 1 h immersion in the alkaline solution. Similar to the study of Moliner et al. [44], the PLA-Sisal bio-composites also show the highest water uptake in alkaline medium (pH 8). This shows the potential of the composites to be treated with alkaline hydrolysis for degradation treatment as it is found to be easily degraded in strong basic solutions [44].

An increase in OPP loadings has led to better biodegradability of the composites due to the presence of higher amount of cellulose content. The high cellulose content (9.19 to 22%) in orange peel has favoured the biodegradation process through hydrolysis and microbial activities [46,47]. Hydrophilic nature of cellulose fibres facilitates the action of water as a carrier for microbes which eventually promotes the enzymatic hydrolysis of cellulose in the composites [29]. Thus, it is believed that chemical composition of the composite plays an important role in determining the microbes and moisture access, which further determines the biodegradability [28,48].

## 5. Conclusions

In this study, orange peel waste transformed into fine powder was incorporated into PLA to develop a biobased, biodegradable polymer. It was proven that OPP has been incorporated into the PLA matrix by the presence of hydroxyl group in the high energy region, and C=C functional group at wavenumber 1608 cm^−1^. Moreover, it was found that addition of OPP into PLA has generally decreased the tensile and modulus, but significant increase of elongation was observed at low loadings of OPP. It was suggested that the OPP which contains high cellulose fibre components modified the toughness and gave a better ductility to the composites. Water contact angle analysis has shown that the hydrophilic OPP significantly modified the surface hydrophobicity of PLA, with higher OPP loadings resulted in rough surface of composites and low surface energy. From soil burial test over a consecutive study of 28 days, it was concluded that the presence of OPP has improved the biodegradability of PLA. Better biodegradation was observed with increasing concentration of OPP.

## Figures and Tables

**Figure 1 polymers-14-04126-f001:**
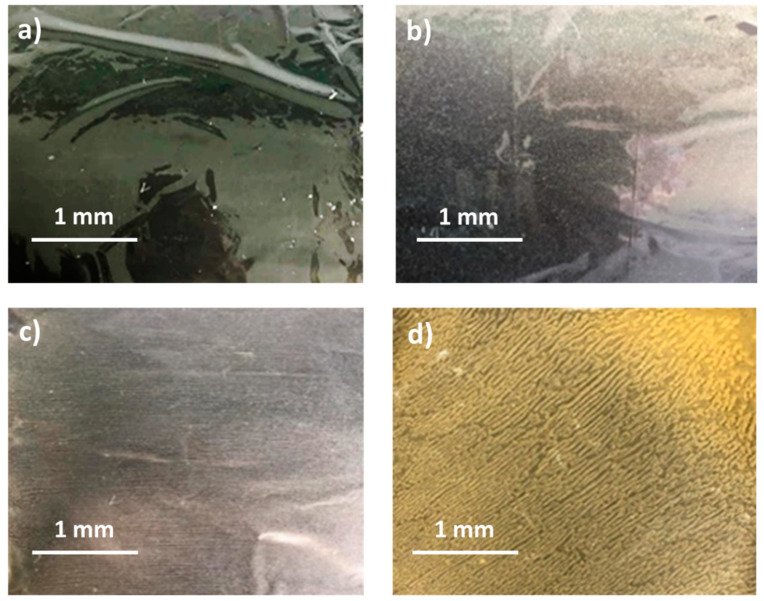
Surface appearance of (**a**) PLA film, (**b**) PLA/10 wt% OPP, (**c**) PLA/20 wt% OPP, (**d**) PLA/40 wt% OPP, and (**e**) PLA/60 wt% OPP composites.

**Figure 2 polymers-14-04126-f002:**
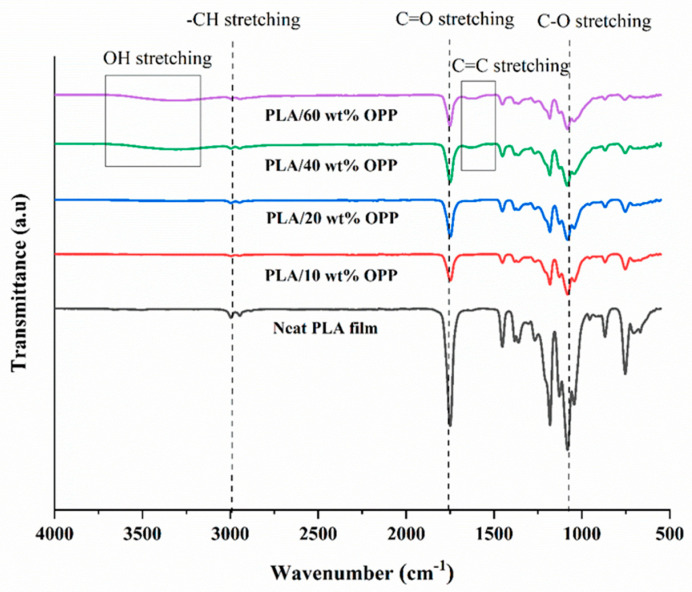
FTIR spectra of samples.

**Figure 3 polymers-14-04126-f003:**
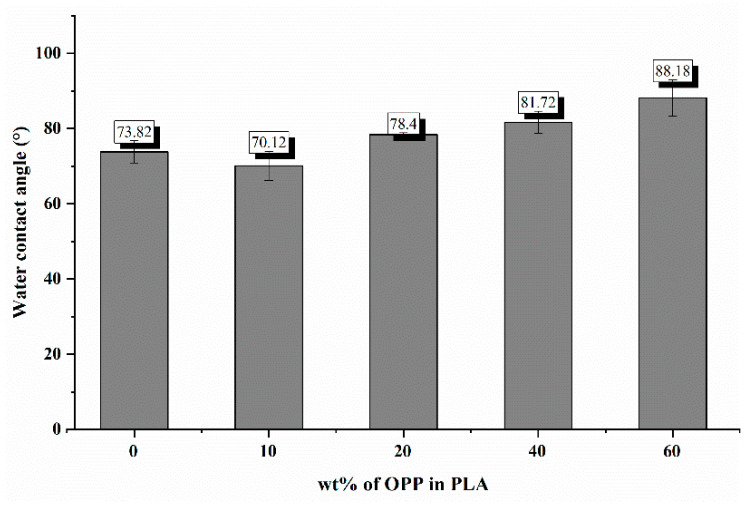
Water contact angle of PLA, PLA/10 wt%, PLA/20 wt%, PLA/40 wt%, and PLA/60 wt% OPP composites.

**Figure 4 polymers-14-04126-f004:**
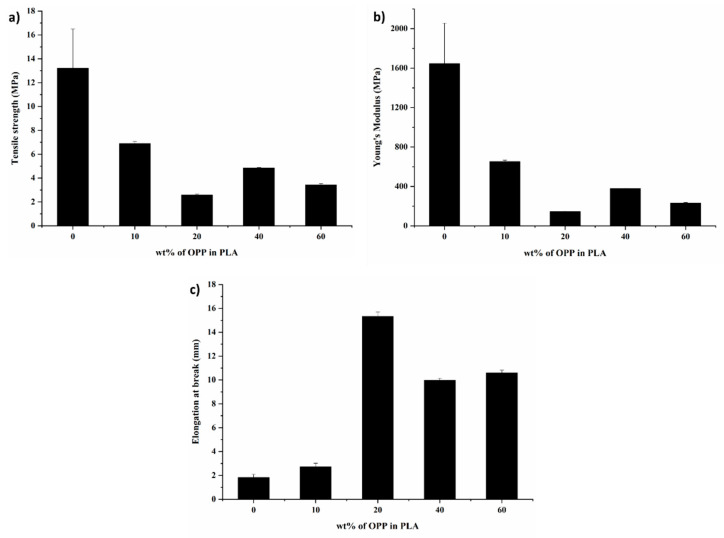
(**a**) Tensile strength, (**b**) Young’s modulus, and (**c**) Elongation at break of PLA, PLA/10 wt%, PLA/20 wt%, PLA/40 wt%, and PLA/60 wt% OPP composites.

**Figure 5 polymers-14-04126-f005:**
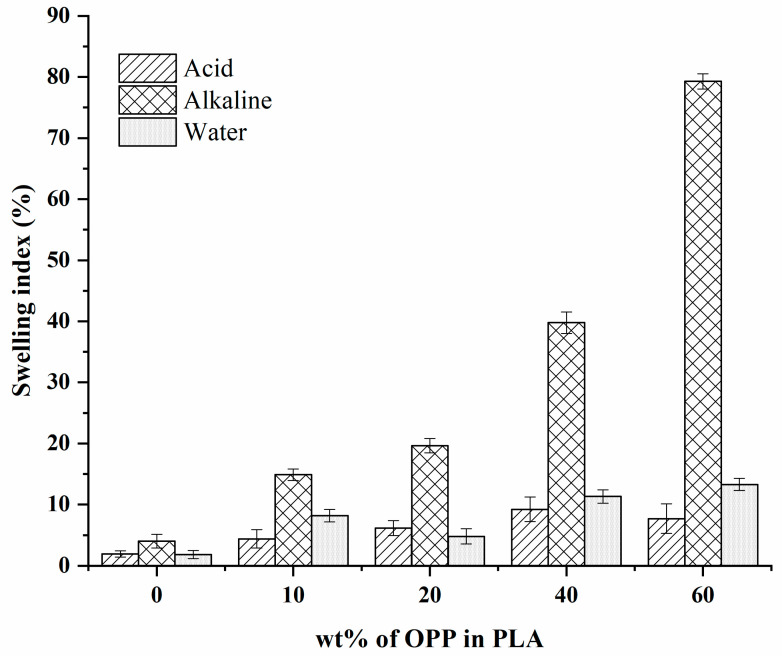
Swelling percentage of PLA, PLA/10 wt%, PLA/20 wt%, PLA/40 wt%, and PLA/60 wt% OPP composites in acid, alkaline and distilled water.

**Figure 6 polymers-14-04126-f006:**
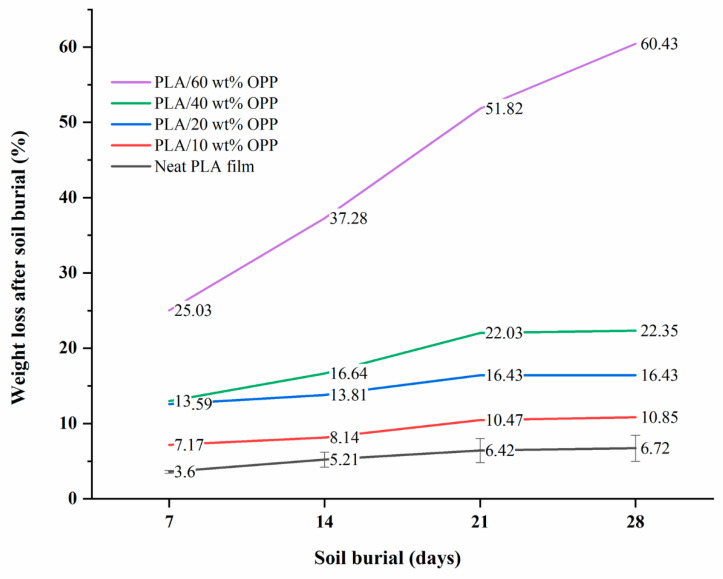
Weight loss percentage of composites over days after soil burial.

**Table 1 polymers-14-04126-t001:** Physical appearance of composites before and after soil burial test.

Composite	Before Soil Burial	After 7 Days	After 14 Days	After 21 Days	After 28 Days
PLA	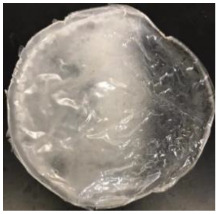	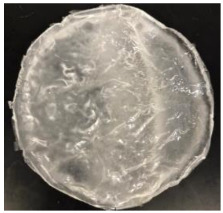	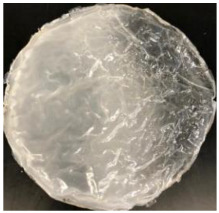	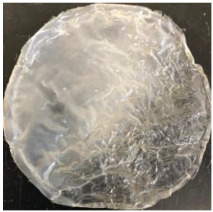	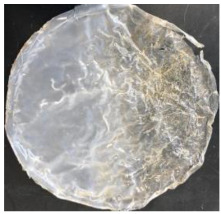
PLA/ 10 wt% OPP	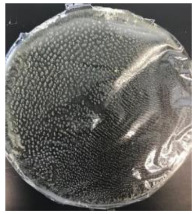	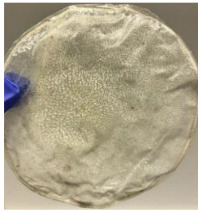	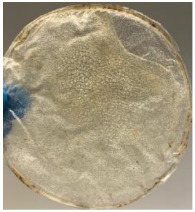	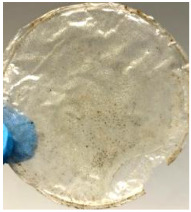	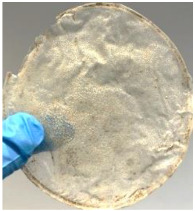
PLA/ 20 wt% OPP	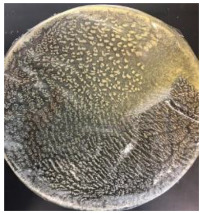	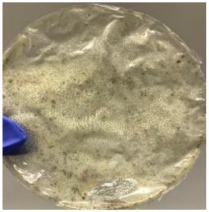	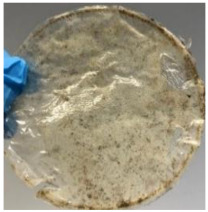	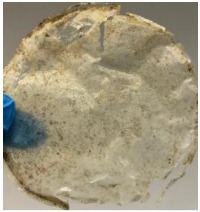	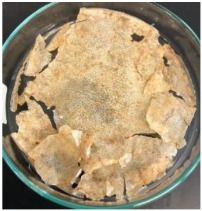
PLA/ 40 wt% OPP	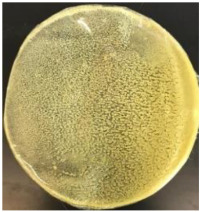	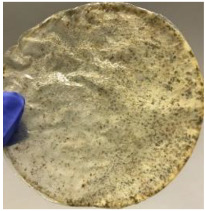	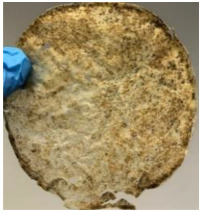	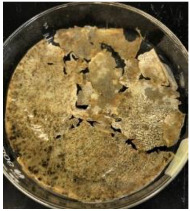	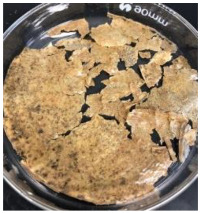
PLA/ 60 wt% OPP	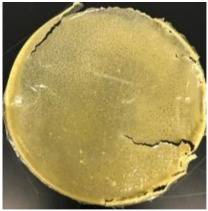	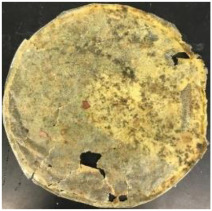	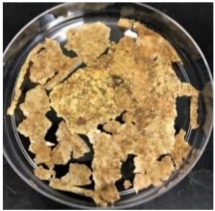	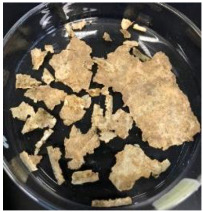	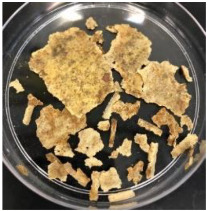

## Data Availability

Not applicable.

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
