# Peer review of "Modification of Poly(lactic acid) with Orange Peel Powder as Biodegradable Composite"

_polymers, 2022, doi:10.3390/polym14194126_

Round 1
Reviewer 1 Report
The manuscript entitled “Modification of Poly(Lactic Acid) with Orange Peel Powder as Biodegradable Composite” investigates the usage of orange peel powder as filler into PLA to improve the biodegradability while maintaining the recorded physical properties. The prepared composite was fully characterized. The topic and given data are promising and match with the scope of Polymers. The manuscript can be accepted after addressing some minor points:
1- In the characterization section, add the name of the microscope used for testing the geometry. Also add the scale-bar in the related fig. 1.
2- Better to collect fig.s 4, 5 and 6 in one summarized fig.
3- Try to make more simplified conclusion to give the main findings directly.
Author Response
We would like to thank the reviewers for constructive comments that help to improve the quality of our papers. We deliberate each of reviewer comment as follows:

Reviewer 2 Report
The manuscript is focused on investigating the incorporation of orange peel powder into Poly(lactic acid), aimed at improving its biodegradability. In general, the manuscript is very well-written, structured, and informative, but still needs some minor improvements before acceptance for publication in the Polymers Journal. Please, see below my comments on your work:
In general, the title (lines 2-3), the abstract (lines 12 to 28), and the keywords (lines 29-30) are relevant to the scope of the manuscript. Overall, the abstract is informative and contains the main findings of the article. I would recommend adding more specific results obtained from your research in the abstract of the manuscript, e.g. about the mechanical properties.
Line 20: although well known, I would recommend providing the full term “Fourier-transform infrared spectroscopy” followed by the common abbreviation FTIR.
Lines 29-30: I’d recommend adding also “biocomposites” to the keywords.
Line 41: in my opinion, it’d be better to replace “ecosystem” with “environment”.
Lines 46-47: please revise the sentence to avoid the repetition of “plastic wastes”.
Lines 49-51: the statements are generally true, but should be supported by relevant references. Please check these examples:
https://doi.org/10.1002/pc.26824
https://doi.org/10.3390/polym13152436
https://doi.org/10.1007/s10853-021-05774-9
https://doi.org/10.3390/polym13050741
https://doi.org/10.1016/j.matpr.2017.02.151
https://doi.org/10.3390/polym13152532
In general, the Introduction section is very well written and provides relevant information on the manuscript topic. The aim of the research is also clearly presented.
Lines 92 and 99: please provide the botanical name of the sweet orange in italics.
Line 101: please explain/justify the duration and temperature used to dry the orange peels.
Line 102: “electronic blender” – maybe “electric blender”?
Line 111: please explain why did you apply these specific orange peel loadings, i.e. 0, 10, 20, 40, and 60 wt%? Is it based on preliminary experiments?
Lines 121-122: please add the standard ASTM D638 in the references of your work.
Lines 123-124: please provide relevant information on the universal testing machine used (company producer, city, country).
Overall, the Materials and Methods sections is clear, descriptive, and very well presented in the necessary details.
Line 221, Figure 8: please revise the figure caption “Days after soil burial (day)” to avoid repetition.
The Results and Discussion sections of the manuscript are very well written. The results obtained from the study are clearly presented, detailed, and properly discussed with relevant research works.
The Conclusion part is well prepared and reflects the main findings of the manuscript.
The References cited are appropriate to the research topic. Most of them are not formatted according to the Journal requirements. Please refer to the Instructions for Authors.
Best regards!
Author Response

(The authors gave the same response as above.)

Reviewer 3 Report
This is the timely effort by the authors to develop the OPP reinforced PLA leading to the characterization of resulting PLA composites. However, there are few suggessions:
1. The novelty is unclear as I can see the similar kind of works in recent years which have also been mentioned in the introduction section.
2. Is the method of making composites standardized?
3. The mechanical properties in terms of tensile strength and young's modulus is found to be lower which puts a big question mark on the research methodology.
4. Are all testing methods especially the measurement method of biodegradability standardized?
5. Discussion section may also be enhanced to justify the weak form of results.
Thanks
Author Response

(The authors gave the same response as above.)
